# Fostering empowerment: Transition from self-help groups to cooperatives in leprosy-affected communities in Nepal

Dilip Shrestha ⓘ *, Bijaya Shrestha ⓘ, Subi Ansari, Sangeet Sharma, Suraj Puri, Ashesh Shakya, Divya Tiwari, Samudra Pandey, Pramila Aryal, Bishnu Dhungana, Jemish Acharya, Shovakhar Kandel, Indra Bahadur Napit

Anandaban Hospital, The Leprosy Mission Nepal, Tika Bhairav, Lalitpur, Kathmandu, Nepal

* dilips@tlmnepal.org

## Abstract

### Introduction

Although leprosy is curable, many affected individuals continue to face disability, stigma, and poverty. In Nepal, sustaining progress after elimination remains challenging, making community-based approaches such as self-care and self-help groups crucial for long-term health and social inclusion. This study investigates how self-help groups and cooperatives contribute to sustaining community-driven leprosy management efforts.

### Methods

For this study only qualitative data are utilized which involves people affected by leprosy, single women, community stakeholders, and facilitators. Conventional content analysis was employed to examine the data, focusing on participants' experiences and perceptions.

### Results

Self-help groups were instrumental in reducing social stigma and promoting social inclusion for marginalized individuals. These groups empowered members by building skills in hygiene, self-care, awareness of gender-based violence, and financial management, enabling them to assert their rights, manage finances, and contribute to household decisions, thereby fostering long-term socio-economic resilience. Transitioning to a cooperative model provided legal recognition and financial security, with leadership and trust being crucial for continued growth.

**Data availability statement:** All data underlying the findings of this study are available within the Supporting Information files (codes and description).

**Funding:** This research was funded by the National Institute for Health Research (NIHR) (NIHR200132) using UK aid from the UK Government to support global health research. The views expressed in this publication are those of the author(s) and not necessarily those of the NIHR or the UK Department of Health and Social Care. The funders had no role in study design, data collection and analysis, decision to publish or preparation of the manuscript. All authors except BD, JA and SK received salary from the funder.

**Competing interests:** The authors have declared that no competing interests exists.

## Conclusions

The transition to cooperatives offers a promising path for sustainable development, ensuring legal recognition and financial stability while reducing stigma and improving societal perceptions towards leprosy, disability, and marginalized communities.

## Author summary

Marginalized groups in Nepal including those affected by leprosy face significant psychological, clinical, social, and economic challenges, exacerbated by stigma and limited access to care and integration opportunities. IMPACT intervention addresses these issues through self-help group and cooperatives. In this study, we evaluated the effectiveness and sustainability of the IMPACT intervention in fostering community engagement and resilience. Our findings revealed that intervention profoundly transformed participants, boosting confidence in health and hygiene practices and reducing social stigma. These groups also facilitated empowerment through skill-building in self-care, gender-based violence awareness, and financial management. Additionally, our findings found transitioning these groups into cooperatives further enhanced their impact by providing legal recognition and financial stability.

## Introduction

The World Health Organization (WHO) defines leprosy as a chronic infectious disease caused by *Mycobacterium leprae*. It primarily impacts the skin, peripheral nerves, the lining of the upper respiratory tract, and the eyes. A combination of medications known as multidrug therapy (MDT) effectively cures leprosy [1]. Although it is curable, early treatment is essential to prevent disabilities. Self-care teaches individuals how to care for their hands and feet to prevent and manage ulcers by inspecting, soaking, scraping, moisturizing the skin, and dressing wounds. The goal is to maintain clean, callus-free, and well-moisturized skin and wounds [2,3]. The relationship between leprosy and poverty is complex and mutually reinforcing. The poor living condition increases the susceptibility of the disease, in turn, the disease contributes to social stigma, discrimination and economic exclusion entrenching poverty. Henceforth, along with the support for physical complications of leprosy, self-help group focuses on the social and economic impacts of the disease. The goal of self-help groups is to support individuals in improving their economic self-sufficiency, enhancing social participation, and advocating for their rights [4,5].

WHO defines elimination of leprosy as a public health problem with less than 1 case of leprosy per 10,000 population at the national level. Nepal successfully eliminated leprosy as a public health concern in December 2009, officially declaring its elimination in 2010 [1]. Despite significant progress in reducing the disease nationwide, sustaining this success and continuing to lower the burden by providing

high-quality leprosy services remain critical challenges [6]. Nepal needs to increase its health sector budget to meet the population's needs, but despite political commitment, the gap between policy formulation and its effective implementation hampers substantial progress [7]. Identifying collaboration with non-state actors as essential has expanded access and involved key stakeholders within the healthcare ecosystem [8].

The Integrated Mobilization of People for Active Community Transformation (IMPACT) initiative was designed to enhance the psychological, clinical, social, and economic outcomes of individuals affected by leprosy. The program consists of self-care and self-help components, offering a multifaceted intervention to address the diverse barriers and enablers to improved health and well-being for marginalized populations. Initially, it focused on supporting leprosy-affected individuals through accurate diagnosis, appropriate treatment, and promoting overall health awareness. Over time, its scope expanded to include other marginalized groups, such as people living with disabilities, those affected by lymphatic filariasis, single women, and people living in extreme poverty, regardless of disease or disability.

The intervention aimed to equip participants with the knowledge and skills needed to improve their quality of life. Medical professionals, along with family members and friends, encouraged participation in group activities, including self-care and income-generating initiatives, to enhance their health, well-being, social integration, and economic status. In longer run, the self- help groups were to transition into a self sustaining cooperative [9]. This research tends to understand the extent do self-care practices, self-help groups, and cooperatives improve the long-term sustainability of community-based leprosy management programs.

## Objectives of the study

The study aims to to explore how self-care practices and the self-help group transition to cooperatives in enhancing the sustainability of community-driven leprosy management programs.

## Method

### Ethics statement

The purpose of the study was explained to potential participants either during a group orientation session or individually. This study received ethical approval from the Nepal Health Research Council (NHRC 444–2020 P). The ethical clearance document is attached with (S1 File). To ensure participants felt comfortable and safe, interviews and FGDs were held in physically accessible, familiar community settings and scheduled at times chosen by participants. Participation was entirely voluntary, and individuals were informed that they could decline to answer any question, take breaks or withdraw from the study at any stage without any consequences. Privacy was maintained throughout. A patient information leaflet and consent form translated in local language and back translated according to the WHO methodology was given to the participants [10]. A written consent was obtained before the interview. The research team was trained in disability-inclusive and trauma-sensitive interviewing, enabling them to respond appropriately to emotional discomfort and provide referral information when needed. Participants were informed that they could skip any question or withdraw at any point without consequence. A supportive environment was maintained by ensuring privacy, using local language, adopting non-judgmental communication, and having same-gender interviewers where preferred. Questions were posed thoughtfully, using appropriate terminology to respect the experiences of participants affected by leprosy, living with disability and those marginalized.

All data collected for this study, including transcripts and field notes, are securely stored in password-protected files on institutional servers at The Leprosy Mission Nepal. In line with Nepal Health Research Council (NHRC) ethical requirements, the data will be preserved for a minimum of five years after publication. Due to the sensitive nature of the information and the risk of participant identification, the full dataset cannot be made publicly available.

## Study design

While the broader study adopted a mixed-methods design integrating both quantitative and qualitative approaches, the present article is based exclusively on the qualitative data used for the analysis. The qualitative study aimed to examine the implementation and sustainability of the intervention. This article involves a deeper exploration of findings from qualitative data, incorporating those from focus group discussions, in-depth interviews, and key informant interviews from participants such as people affected by leprosy, single women, community stakeholders and facilitators. The study adhered to the guidelines of the Consolidated criteria for reporting qualitative research (COREQ), a 32-item checklist that promotes comprehensive reporting of qualitative research by addressing key areas such as the research team's role, reflexivity, study design, data analysis, and findings. It also covers details like the research topic, study goals, objectives, and the context of the study, along with sampling methods, participant recruitment, and ethical considerations [11].

## Study setting

The study was conducted in three program districts of Lumbini Province (Rupandehi, Kapilvastu and Nawalparasi-west). Two municipalities were selected from each district. Rupandehi (Devdaha Municipality and Rohoni Rural Municipality), Kapilbastu (Banganga Municipality and Shivaraj Municipality), and Nawalparasi-west (Sunwal Municipality and Sarawal Rural Municipality). Fig 1

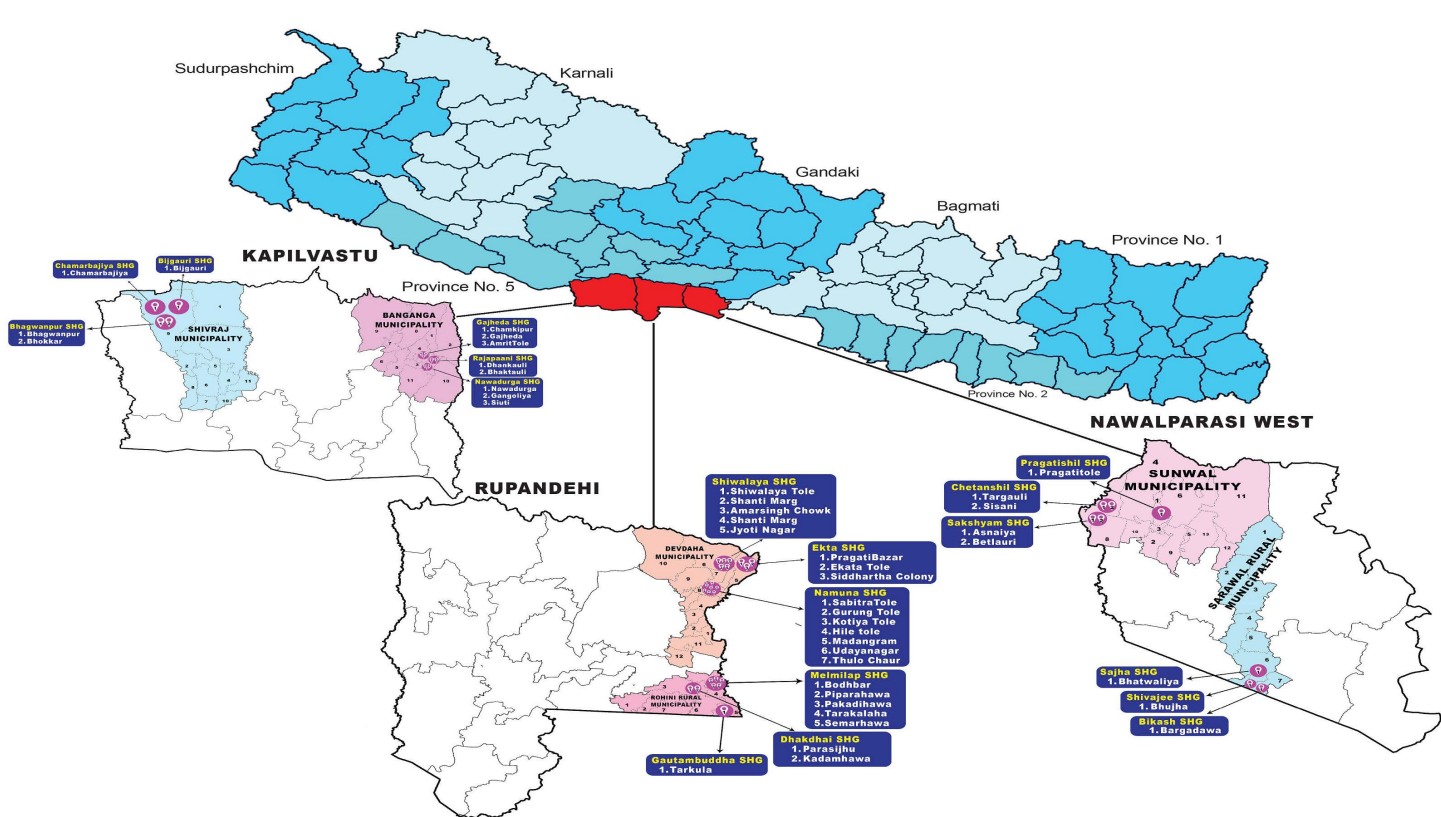

**Fig 1. The study site selected for this study was Lumbini Province.** As shown in figure, three districts; Kapilvastu, Rupandehi and Nawalparasi west were selected. From those selected districts, two municipalities were selected for the research. The figure was constructed manually and used in the protocol [9].

A comprehensive understanding of the administrative structure of Nepal and the implementation details of the IMPACT intervention can be found in another article [9].

Within each cluster, the IMPACT intervention formed a group of about 25 eligible individuals, including those with disabilities, affected by leprosy, or marginalized people. These members were typically adults aged 18 years and above from separate households. This shows the diversity of the individuals enrolled for research.

In the six municipalities, a total of 36 self-help groups were formed during the project phase, and participants were selected from eighteen self-help groups. The self-help groups (SHG) were initially established to provide support and empowerment to marginalized individuals, including those affected by leprosy, disabilities, lymphatic filariasis, and those from economically disadvantaged backgrounds, particularly single women. These groups were later transitioned into cooperatives, aiming to enhance their effectiveness and create a more sustainable and inclusive environment for their members. The focus group discussion (FGD) and in-depth interview (IDI) participants in this study, who were also members of the self-help group, were encouraged by doctors and volunteers to join the group, their friends and family referred a few.

### Sampling and participants

The participants were selected from 18 SHGs. For the focus group discussion, a random sampling approach was opted to sample the study participants. A random sample (N = 9) of groups stratified by facilitator was selected. A random selection of 6–10 group members were invited to participate in the discussion without the facilitator present. Random selection helped capture a broader range of experiences- including those who may be less confident, less visible, or less involved in group activities- thereby strengthening the validity and balance of data. The participants were included from the self-help group members and those who did not consent for the interview were not included in the study. With the agreement of the study participants, the meeting took place at the specified venue following the scheduled facilitated session. During the session, participants were asked about their daily experiences and relationships within the community. The content of the discussion was observed, and the interactions between participants were noted. Altogether nine FGDs were conducted and 80 participants participated in the FGDs. (Table 1) Participants of FGDs were encouraged to share their perceptions and impressions on their experience of the intervention.

**Table 1. Participants characteristics.**

| S.N. | Type of interviews | Sex | |
|------|--------------------|-----|-----|
| | | M | F |
| 1. | FGD | | |
| | Total | 9 | |
| 2. | In-depth interviews (IDI) | | |
| | Leprosy affected | 4 | 5 |
| | People living with disability (PLWD) | 2 | 2 |
| | Single women | | 2 |
| | Marginalized | 2 | |
| | Family Members | | 1 |
| | Total | 8 | 10 |
| 2. | Key Informant Interview (KII) | | |
| | Core Facilitator | 2 | 1 |
| | Local Facilitator | 2 | 6 |
| | Broader Community | 12 | |
| | Total | 14 | 6 |

The IDIs were conducted to gain insight into the intervention's impact on individual lived experiences with two randomly selected group members from each group (n = 9), stratified by gender (one male and one female). A total of 18 IDIs were conducted, and the recruitment took place at the end of the group interview, using a lottery method for transparency. (Table 2) During the in-depth interview, participants led the researcher to their homes, workplaces, and other frequently visited locations, discussing how their health conditions affected their daily lives.

Key Informant Interviews (KII) were conducted with individuals from the broader local community, including community leaders, religious leaders and healthcare professionals, to gather their attitudes toward people with leprosy and people with disabilities. A purposive sampling was opted, and 12 individuals were invited to participate in the interview that took place as per the convenience of the participants. (Table 3) The sampling of community members followed the initial analysis of participant interviews, as these provided insights into which community members might mediate the intervention. The interviews explored participants' experiences, knowledge, and views on leprosy and other diseases that cause ulcers, interventions for these conditions, barriers and facilitators to community support for affected individuals, experiences of stigma, and the perceived impact of IMPACT on both participants and the broader community.

Interviews were also conducted with eight local facilitators (LFA) (Table 4) and three core facilitators (CFA) of the self-help groups (Table 5). The interviews explored academic background, professional experiences, roles and responsibilities in the project, knowledge, and views on leprosy, details on case findings and screening of leprosy, challenges and barriers experienced during program implementation, social, economic and health reform as an impact of IMPACT intervention.

Further, the researchers used a checklist based on the framework to record behaviour change techniques during the group meeting. The researcher observed how different facilitators/group leaders interpreted their role, how they interacted with the group and how group members responded and recorded as field-notes. These observations helped understand the fidelity with which the principles of the intervention are applied and any local deviations.

**Table 2. Qualitative in-depth Interview.**

| Participant ID number | Education | Age | Gender | Marital status | Position | Comments |
|---|---|---|---|---|---|---|
| 1 | Grade 10 | 42 | F | Married | Member | Leprosy |
| 2 | Grade 10 | 25 | F | Unmarried | Group Leader | Leprosy |
| 3 | Bachelor | 43 | M | Married | Member | Leprosy |
| 4 | Grade 8 | 52 | M | Married | Secretary | Disable |
| 5 | Grade 3 | 67 | F | Married | Member | Leprosy |
| 6 | Grade 6 | 34 | M | Married | Member | Disable |
| 7 | Literate | 44 | F | Married | Member | Member |
| 8 | Grade 2 | 52 | F | Widowed | Member | Single woman |
| 9 | Illiterate | 66 | M | Unmarried | Member | Leprosy |
| 10 | Grade 2 | 47 | M | Married | Member | Leprosy |
| 11 | Illiterate | 57 | F | Married | Member | Single woman |
| 12 | Grade 12 | 23 | M | Unmarried | Member | Disable |
| 13 | Grade 10 | 34 | F | Married | Member | Marginalized |
| 14 | Illiterate | 40 | M | Married | Member | Leprosy affected |
| 15 | Grade 8 | 36 | M | Married | Member | Disable |
| 16 | Grade 8 | 41 | F | Married | Chairperson | Leprosy affected |
| 17 | Illiterate | 53 | M | Married | Member | Leprosy |
| 18 | Illiterate | 55 | F | Married | Member | Marginalized |

**Table 3. Qualitative interview broader community.**

| Participants ID Number | Education | Position | Age | Gender | Marital status | Address |
|---|---|---|---|---|---|---|
| 1 | Sr. AHW | Health Officer | 52 | M | Married | Devdajha Municipality-07 |
| 2 | B.com, CMA | AHW | 55 | M | Married | Bangnaga Municipality-04 |
| 3 | CMA | Sr. AHW Officer | 54 | M | Married | Sunuwal Municipality-01 |
| 4 | Grade 10 | Ward Chairperson | 61 | M | Married | Sunuwal municipality-01 |
| 5 | Grade 10 | Ward Chairperson | 44 | M | Married | Shivraj Municipality-09 |
| 6 | Grade 10 | Ward Chairperson | 60 | M | Married | Rohini rural municipality-03 |
| 7 | Grade 10 | Ward Chairperson | 42 | M | Married | Sarawal rural municipality-07 |
| 8 | ISC | Agriculture Officer | 52 | M | Married | Banganga Municipality-04 |
| 9 | ISC | Agriculture Officer | 58 | M | Married | Devdajha municipality |
| 10 | Grade 12 | Church Leader | 44 | M | Married | Bangnaga Municipality-04 |
| 11 | Masters | Principal | 44 | M | Married | Shivraj Municipality-09 |
| 12 | Educational management | No specified | 53 | M | Married | Banganga Municipality-04 |

**Table 4. Local facilitator.**

| Participant Recording Number | Municipality | Education | Position | Age | Gender |
|---|---|---|---|---|---|
| 1 | Devdaha Facilitator | Bachelor | Facilitator | 34 | F |
| 2 | Sarawal Facilitator | Grade 12 | Facilitator | 38 | M |
| 3 | Rohini Facilitator | Bachelor | Facilitator/GL | 20 | M |
| 4 | Sunuwal Facilitator | Grade 10 | Facilitator | 40 | F |
| 5 | Sunuwal Facilitator | Grade 6 | Facilitator/GL | 32 | F |
| 6 | Sunuwal Facilitator | Grade 12 | Facilitator/left | 39 | F |
| 7 | Shivraj Facilitator | Grade 10 | Facilitator/GL | 22 | F |
| 8 | Banganga Facilitator | Grade 10 | Facilitator | 21 | F |

**Table 5. Core facilitator.**

| Participants ID Number | Municipality | Education | Position | Age | Gender |
|---|---|---|---|---|---|
| 1 | Rupandehi | Master in Health (MEd) | District Officer | 39 | F |
| 3 | Kapilvastu | Social science and political science | Health and DRR coordinator | 56 | M |
| 2 | Bhairahwa | Sociology | Livelihood Coordinator | 42 | M |

## Data collection

The first author (DS) and research officers (SA, SS, AS, SP[1], DT, SP[2], PA) conducted in-person interviews. The FGD, key informant interview (KII) was conducted by the first author. DS is a project Hub Manager and currently a DrPH Scholar at Mahidol University, previously working on the project [9], which was led by co-author Indra Bahadur Napit (IBN), who has extensive experience in the field of leprosy and is currently working at The Leprosy Mission Nepal (TLMN). Fieldwork began in July 2021, and data analysis was completed by May 2024. The piloting of the topic guide was conducted in a similar community and group of up to 10 individuals to ensure consistency across sessions. Formal interviews and group discussions were conducted only once as per the ethics using a topic guide, with each session lasting between 30–90 minutes. Audio recorders were used to record the interviews and discussions with prior consent. Since this research was part of an evaluation, the program team was stationed at the field site, helping to build relationships and rapport with the participants.

## Data saturation

Data saturation was achieved through the triangulation of multiple data sources, including interviews with diverse participants. The interview contents were reviewed and categorised across project benficiaries, community-based stakeholders and project facilitators on 'Self-care', 'SHG', 'Cooperatives', and 'Sustainability'. Interviews were conducted until data saturation was attained, defined as the point at which additional data failed to generate new codes or add meaningful variation to the existing themes.

## Data analysis

In this study, we aimed to analyze the data from interviews conducted during the project period to achieve the research objectives. A conventional content analysis was employed, which involved analyzing the textual data and interpreting how participants shared their experiences [12]. Throughout the research process, the researcher remained attentive to understanding the participants' experiences, expressions, and perceptions of reality. The transcripts were coded by BS, and the results were discussed with DS, IBN, BS. In cases of any discrepancy, IBN, BS and DS resolved the issues.

The recorded interviews were first transcribed into English, and the transcripts were checked randomly against the audio recordings to ensure accuracy. During the initial stages of analysis, the researchers first listened to the recordings to gather preliminary insights. Afterwards, the transcripts were scanned and then thoroughly read. The entire narrative data was read repeatedly to gain immersion and form an overall perception of the content while taking notes on what the text was conveying and the main impression of the situation (first credibility criterion). The data was then examined word by word, with the most descriptive parts of the text about key concepts or thoughts highlighted, leading to the development of meaning units. These meaning units were shortened by selecting exact words from the underlined text that captured the participants' key thoughts or experiences, resulting in condensed meaning units. Finally, these condensed meaning units were labelled with one or two words directly derived from the text, forming the development of codes. The Taguette software was used to organize and identify patterns in the data set and validate the researchers' analysis [13]. Upon completion of the coding, the excerpts were exported on an Excel sheet. The data was structurally managed on the Excel sheet, where the excerpts were aligned across the codes and type of the study participants. Additional columns were added for the excerpts summary and the researcher's reflection. Narratives were extracted from the Excel sheet to complement the findings.

## Clarity of major themes

The themes were clarified based on data analysis, where emerging patterns were consistently identified and refined. By continually comparing data across participants and revisiting initial codes, the key themes became distinct and well-defined, ensuring a clear understanding of the core findings.

## Results

This section presents the outcomes of the leprosy elimination efforts and healthcare interventions aimed at marginalized communities in Nepal. The data highlights both the successes and challenges the IMPACT program faces, focusing on disease prevalence, accessibility of healthcare services, and improvements in community well-being. Further, it reflects the effectiveness of targeted interventions while uncovering gaps that require further attention to ensure sustained progress in healthcare equity, disease control and socio-economic development. Altogether nine FGDs were conducted comprising 80 participants, which included project beneficiaries representing a diverse group. For IDI, 18 participants were selected randomly from the FGD participants. Additionally, for KII, 20 participants were interviewed, consisting of facilitators and community-based stakeholders (Table 1). The details of participants enrolled are provided in S3 File. Findings are presented below under four key themes: 1. Self-care, 2. SHG, 3. Cooperatives, and 4. Sustainability.

## 1. Self-care

**Self-care and hygiene practice.** The participants from IDI and FGD highlighted the transformative impact of the self-help group on their behaviour, confidence, and social integration. Most of the participants emphasised the benefits of self-care practices taught by the group, such as washing hands regularly and boiling water before drinking. The majority had received basic tools like soap, sanitizers, buckets, mugs, and tubs for soaking feet and hands, which is essential in preventing them from injuries, in addition they also received customised shoes, basins for wound care, and teaching self-care practices.

*"I soak my feet in cold water for some time and apply mustard oil. This routine has greatly benefited me; my leg has healed a lot and improved significantly. It used to pain a lot, but now the pain has subsided."*

-IDI, P05, Female, Leprosy affected

Facilitators and leaders aligned with the participants regarding the positive trajectory of self-care in addressing leprosy through medical treatment and community awareness.

## 2. Self help group

**Social support and stigma.** The self-care group became a safe space for members to share their lived experiences, discussing struggles and seeking support from one another. In some cases, potential members did not attend, citing job priorities. Additionally, some members from affluent families were reluctant to join the disease-specific group. In extreme cases, individuals hid their diagnoses, even from their own families, to avoid social exclusion and fear of ostracism. Local stakeholders recounted instances where families isolated leprosy-affected members, creating separate kitchens or even relocating due to societal pressure. Over time, the group evolved into a self-help group, expanding to include people with disabilities, individuals affected by lymphatic filariasis, single women, and individuals from economically disadvantaged backgrounds, alongside those affected by leprosy. This expansion fostered a supportive and inclusive environment, enhancing the groups overall effectiveness.

*"I joined the group because I am a single woman from an economically backward class. This group includes single women, people affected by leprosy, disabled individuals, and those who are economically and socially marginalized."*

-FGD 02, P2, Female

*"Being associated with the group gives us hope that we will accomplish something together. In the past, we were not part of any group and struggled on our own.."*

-FGD 02, P8, Female

Despite initial challenges, the group was eventually formed and became crucial in changing attitudes towards leprosy. Participants from FGD noted that the trainings and awareness sessions through the group clarified misconceptions about leprosy, particularly the belief that it is a communicable disease and was treatable. Gradually, there was a sense of community support, where members encouraged one another to seek treatment, and the community's fear towards those people with leprosy significantly declined.

*"In the past, there were terms like 'Kod' or 'Kodiya' (derogatory terms used for leprosy patients to demean them), and people believed that those with such infections should not be touched, that we should not sit with them or eat with them. These ideas were prevalent and commonly heard before, but now this disease is understood to be like any other, and it can be cured with proper treatment."*

-KII, Broader 08, Male, Agriculture Officer

The group members, facilitators, and community stakeholders aligned their perceptions, contrasting past challenges with current improvements, underscoring the groups effectiveness. Participants affected by leprosy mentioned the existence of minimal to no discrimination in their communities and the reduction of stigma.

*"People were not aware of this disease, but now they are aware of it and its treatment, and the discrimination has stopped."*

-FGD 04, P6, Female

*"Before there was some form of discrimination, but after the group was established, people realized leprosy is not bad or life-threatening, it can be cured with the help of medication, and gradually people viewpoint started to change."*

-FGD 05, Male

Some facilitators mentioned the geographical and socio-cultural factors posed challenges in managing the group dynamics.

*"The reason for the difficulty was the caste system. Now… in the caste … (trying to find words to express the statement). The caste system in Nepal still exists. The so-called upper castes don't want to keep the so-called lower castes in the group. They always have the mentality of being the leader. There are fifteen members, of which fourteen are Dalits and Janajati (tribals), right? Even though there is one Brahmin or Kshatriya, he/she desires to be the Chairman."*

-KII, CFA 02, Male, Coordinator

While the facilitators initially expressed frustration over initial challenges in forming the group and implementing group activities, they also conveyed a sense of perseverance. They described their efforts to convince people to join these groups despite facing significant resistance, emphasizing a personal commitment to social change. Furthermore, a CFA appraised that the group not only raised awareness but also fostered greater social integration by encouraging people to eat and train together; the group has helped break down social barriers between leprosy-affected individuals and the rest of the community.

*"I have seen a drastic change in the attitude of the people. When I was working as an FCHV, someone I knew was diagnosed with leprosy.It was a hush-hush situation and patient were secretly taking medication. They were apprehensive, thinking that if others found out, they would be ridiculed and treated harshly. However, after the self-help groups were formed, people now share the information openly among the villagers. They say that even if you get the disease, if you take the medicine, you will be fully cured. There has been a great change in attitude. Previously, even the health post worker would just give us the names and ask us to keep it to ourselves, contacting the patient with caution. Now, there is no problem at all."*

-KII, LFA 07, Female, Facilitator

**Training and skill development.** With the expansion of the self-help group, participants reported a comprehensive array of trainings offered, which included training on Capacity Building, Psychosocial Support, Emergency Response, Debate and Advocacy, Leprosy Awareness, Water, Sanitation, and Hygiene (WASH), Disaster Management, Accounting, Self-Care, Gender-Based Violence, Sanitary Pad Production, Mental Health, Cooperative Management, Health Training, and Equity and Equality.

*"A facilitator came and provided us with training. We learned that different colored identification cards blue, white, red, and yellow are distributed for types of disabilities. We were also taught the importance of maintaining cleanliness and proper hygiene, and how we can care for our wounds by softening them with water and using the appropriate creams to avoid complication.*

-FGD 01, P7, Female

Meanwhile, facilitators emphasized providing more training, especially skill-based training that makes members economically self-sufficient. Likewise, training in marketing and economic management is needed as people are struggling with selling their product, indicating a gap in their ability to complete the business cycle. Furthermore, the community-based stakeholders highlighted the need for the active involvement of local bodies to ensure the continuity and sustainability of the activities even after the external project ends.

*"Interviewer: For the continuation of the activities and setups that have been undertaken, what do you think should be done? What are your thoughts, Sir?*

*Interviewee: It should coordinate with the local bodies. It should present to the local bodies all the progress that has been made so far.*

*Interviewer: Reports?*

*Interviewee: Yes, the progress should be reported. It should be communicated that this is the progress we have made up to now, and it should be continued and sustained. If it is not sustained, then all the efforts will be in vain. It is important to make the local bodies understand this. If they grasp this, the program will be taken up at the local level, as it will be handed over to them. The key is to make them understand the importance."*

-KII, Broader 02, Male, Health Officer

As mentioned by the group facilitators, members expected the group to distribute money like other projects and were disappointed to find it was focused on providing training and skills to leprosy-affected people. Believing the organization would also eventually leave, they dismissed it as a waste of time and left the group.

Some members felt a blanket approach was opted to train the members, and the need for more specific skill-based training for certain individuals was not considered. The communication of training information was a concern; some members felt that training opportunities were not equally accessible, and that information was not shared adequately. For example, participants expressed frustration over the lack of adequate training for themselves.

*"Uhm… the thing I haven't liked is that, if there is a training, then everyone needs to be informed. I don't know whether there is a quota for a certain number of people or what the system is. This information should be circulated to everyone. Let us suppose that there are thirty people, and only fifteen people attend the training. The remaining fifteen should be given training the next time."*

-IDI, P15, Male, PLWD

**Role of self-help groups.** Support from SHGs and individual members was crucial for recovery and regaining functionality as the groups not only facilitated access to appropriate medical care but also provided emotional and financial support, alleviating the burden of treatment expenses and travel costs. Many participants highlighted how the group increased their understanding of leprosy and personal health. Participants gained significant knowledge about leprosy, including how to detect the disease (e.g., skin patches), the importance of early diagnosis, and the effectiveness of medication in curing the disease,

which they had not understood prior to joining the group. Most participants gained crucial knowledge about leprosy through SHGs. For instance, a participant noted that chronic conditions like leprosy could lead to permanent disabilities if not managed well. They now learned self-care, hygiene, and cleanliness to prevent leprosy and other diseases.

*"As someone affected by leprosy, I contracted the disease when I was around eleven or twelve years old, but I didn't realize it at the time. I noticed a patch on my arm, which made me anxious as I wondered how it could be cured. Questions flooded my mind: What can be done? Where can I get medicine? Is it free, or will I have to pay for it? I asked people for advice on what to do and where to seek treatment. They told me that free medicine was available at the health post, and was advised to take it regularly for a year. I followed the treatment diligently for six months."*

-IDI, P02, Female, Leprosy affected.

Interestingly, a participant showed resilience, which she had built overtime against future challenges. After being associated with the group, she understood the importance of managing savings and allocating resources to essential needs indicating responsible behavior and awareness of long-term planning.

*"I have saved some of the money, and used the rest to buy clothes and other things I needed. I buy medicine for myself, and I have purchased insurance for One and a half lakh, and I annually deposit the installments."*

-IDI, P02, Female, Leprosy affected

Most stakeholders and participants had observed reduced incidence of common colds and fevers after adopting better hygiene practices. As stated by some stakeholders, approximately half of the community had begun boiling water before consumption, improving overall public health. Participants shared that the knowledge gained from the group was not only for personal benefit but was also disseminated throughout their communities. There had been an emphasis on self-care and regular screening, with monthly check-ups and an increased focus on maintaining cleanliness.

*"Previously, we often suffered from common colds, coughs, and frequent fevers. However, after we started washing our hands with soap, these problems have significantly reduced. By using masks and hand sanitizers, we've been able to avoid these illnesses entirely."*

-FGD 02, P6, Female

With the support of SHGs, several participants had registered for disability cards, but some reported not receiving any tangible benefits despite having a registered card. There was a clear discrepancy in how distinct categories of disabilities had received support, with those in category 'D' receiving no benefits [14].

**Empowerment and behavioural change.** Many of the participants from FGD and IDI stated an increase in their confidence to speak publicly and engage in social settings. They believed their participation in various trainings contributed to enhanced productivity and confidence. The various training and financial support helped to build businesses, manage finances, and engage more effectively in their communities. They witnessed the transition from dependency to self-reliance through their businesses and economic activities and attributed it to the support received from the group. Their enhanced financial situation was reflected in their ability to manage savings and invest in different vocations.

*"This SHG has enhanced our confidence. Now we can talk to others without any fear and hesitation. We are getting different trainings time to time. Our knowledge level has been raised. Due to all these things, we are very much happy."*

-IDI, P01, Female, Leprosy affected

Many viewed the group as a supportive family where they could share problems and seek help. The sense of belonging and mutual support became a significant factor in improving their overall well-being. Some participants highlighted how they previously experienced social exclusion due to their leprosy or difficult circumstances but found solace in the group. The emotional bonding and collective problem-solving within the group made participants feel less isolated and more supported. For instance, a participant reflected on the shift in self-perception from feeling "poor" to walking with "pride and dignity".

Most of the participants were grateful to the group as they had gained knowledge about their rights and available resources, including obtaining citizenship certificates and other essential documents. Before joining the group, many women felt socially and economically dependent, but now they expressed increased autonomy and a greater ability to contribute to household and community decision-making. Single women became more assertive and conscious of their rights, feeling empowered to stand up against discrimination and seek legal recourse when necessary.

*"Earlier we couldn't speak about anything. Earlier society would judge and say things like, 'gosh this household has someone with such debilitating condition, how will they manage? But now they have changed. Personally, I feel that I can walk with my head held high and can speak with confidence and conviction in front of the society. I can take care of my family and guide them to a good life with ease."*

-IDI, P07, Female, Member

While the majority of the members felt empowered, few, especially the older population weren't confident to participate in the group activities and to try out new endeavours.

*"I have no education, so it's a struggle to understand things. Life has been difficult, I have aged, had I been younger than things would have been different. I could have learned things more easily. We are mostly engaged in agriculture and farming and rearing animals."*

-IDI, P05, Female, Leprosy affected

With the success in changing attitudes towards leprosy the group members started taking initiative and caring for themselves, indicating growing self-confidence and self-esteem. Interestingly, individuals who recovered from leprosy were helping others in the community and were now following the example of those who helped them.

*"For the individuals affected, we conduct trainings every month. If they have symptoms, they come to me for consultation. I inform them that they can get the medicine free of charge at the health post. I advise them to go to Bhagwanpur (nearby health post), get checked there, and take the regular medication provided. Some girls, who feel shy discussing their condition with doctors, come to me for support. Once I accompany them, they feel confident to continue on their own. This has led many start regular medication and recover from the disease."*

-IDI, P09, Male, Leprosy affected

Most of the facilitators highlighted how members previously shy gained confidence in speaking publicly. Most members felt more empowered to speak, engage with outsiders, and participate in discussions. The stakeholders described similar observation with a tremendous change in group members' lives. Initially, those hesitant to join the group and unable to introduce themselves confidently were now competent, empowered, and able to express themselves. Most members were now aware of their rights and available resources and were comfortable seeking aid from various forums.

*"The feeling of 'I too can do something' has developed. In the past, we noticed disappointment, but now people say, 'Why can't I do it?' and 'I am not weak.' Their confidence seems to have increased. Additionally, the capacity to stand up and speak in front of others has developed."*

-KII, Broader 08, Male, Agriculture Officer

**Access to medical care and leprosy awareness.** The group leaders highlighted the provision of accurate diagnosis and appropriate treatment throughout the group. The leaders also underscored the value of health awareness training, which included education on disease prevention and general hygiene practices. Additionally, they highlighted that group members were able to access free treatment for severe leprosy cases, including referrals to Anandaban Hospital (A tertiary level hospital for leprosy care) in Kathmandu for specialized care.

Participants from IDI and FGD shared multifaceted experiences with leprosy management that influenced their decision to join the group. For instance, some participants had encountered challenges in obtaining a proper diagnosis and treatment. Some had initially sought help from local doctors who misdiagnosed or failed to recognize the symptoms of leprosy, leading to delays in diagnosis. A few participants had travelled significant distances to reach health centres. Additionally, the psychological toll of leprosy was evident, with fear, embarrassment, and hopelessness, particularly upon receiving their diagnosis. One participant described crying after being diagnosed, initially believing she was doomed, until reassured by family members that the disease was curable.

*"Why wouldn't I panic and feel fearful? I genuinely thought I might die from this disease. The skin around the infected area became thick and lost all sensation, while the rest of my skin remained normal, with full sensation. It's hard not to panic when you experience something like this."*

-FGD 04, P4, Female

Some participants highlighted the availability of free medicine for leprosy and its positive impact on their treatment-seeking behaviour. Many were treated at local health posts or hospitals, while some had to travel to various locations for diagnosis and treatment, indicating geographic disparities in healthcare access. Despite receiving treatment, some participants continued to live with physical disabilities or side effects of leprosy. These physical changes affected their ability to work and led to significant emotional distress, including anxiety and fear of permanent disability.

*" When I was around 12 or 13 years old, I had swelling in my hands, but I didn't know the disease or how to treat it. I grew up in a rural village, and it wasn't until I moved here that I learned what was the cause. I asked for help, and a female community health volunteer (FCHV) in our village advised me to visit the health post, where I could get free medicine to cure the disease. I began my treatment in 2072 BS (2015 AD), and after taking the medicine, I was cured. We also received travel allowances, about NPR 1000 ($7.50), to help with expenses, and eventually, I recovered completely."*

-FGD 03, P1, Female

Almost all the stakeholders shared similar observations and emphasized how SHGs facilitated access to healthcare and supported participants with resources such as wound-cleaning materials, soap, and utensils to help them manage leprosy-related disabilities. Most stakeholders also mentioned the distribution of resources such as prosthetic limbs, special shoes, wheelchairs, crutches, and self-care training, to enhance the mobility of people living with disabilities.

### 3. Cooperatives

**Economic empowerment and cooperative development.** Most participants expressed a strong desire to acquire new skills and initiate businesses, hoping to support their families better and manage daily expenses. In response, the group scope extended beyond healthcare to offer loans at a minimal interest rate. These loans, while more accessible and flexible than traditional bank loans, business commenced through the loans had varying impacts on members.

*"I took money from the organisation, and I started fish business. I have been making good profit from the business."*

-IDI, P06, Male, PLWD

*"Being associated with the group I was able to access a loan, start a poultry business and opportunity to own a shop. If you seek for loan in the general market, it is difficult."*

-IDI, P07, Female, Member

To foster economic stability and strengthen community support, the self-help group transitioned into a cooperative model. This transition encouraged members to engage in savings and expanded membership from its initial focus on individuals with leprosy, People living with Disability, affected by lymphatic filariasis and single women, to include a broader community people. The shift to a cooperative was clearly communicated to group leaders, facilitators, and members alike. Both facilitators and leaders recognized the necessity of this transition for achieving legal authorization and ensuring long-term sustainability. The group leaders were confident that the cooperative model would provide substantial benefits compared to traditional self-help groups, such as a formal legal framework, improved financial and economic stability, greater consistency and accountability, and more reliable access to long-term funding.

*"A SHG is an informal entity that can only sustain itself as long as its intentions remain good. However, without legal backing, there is always a risk of dissolution. When dealing with loans, savings, business startups, and fund collection, it is challenging to manage these aspects through an informal body... a cooperative has a board, management team, and employees, which contribute to the long-term sustainability. Cooperatives maintain a continuous flow of funds, which supports ongoing attention and activity.*

-KII, CFA 03, Male, Facilitator

However, logistical difficulties arose, such as the lack of a proper meeting space and difficulties in organizing meetings, including difficulty in reaching out to the leadership were highlighted. Lack of cooperation from the leadership also highlighted the issues and affected some of the cooperatives. The group adjusted by understanding their situation and worked quickly to accommodate their schedules.

*"We didn't have much discussion because secretaries were always in a rush to get back to the school. They would tell me, 'Madam, we need to finish this quickly, else we will be reprimanded at school and could lose our job'."*

-KII, LFA 01, Female, Facilitator

Most of the participants from IDIs repeatedly mentioned 'unity' as a crucial factor for long-term sustainability. They believed that collective efforts and a shared sense of responsibility among the members would enable the cooperative to continue and operate smoothly, even in the absence of supporting organization. This includes managing loans and interest payments, as well as reinvesting the funds to benefit all members. Selecting reliable members to oversee loan management and avoiding the inclusion of unreliable member were considered essential for the cooperative success and

sustainability. Additionally, some of the participants suggested extending membership, advancing farming practices, and enhancing the member's financial contribution for the sustainability.

> "Interviewer: Do you think the cooperative will remain functional in the future?
>
> Interviewee: Yes, it will continue
>
> Interviewer: How do you think it will operate?
>
> Interviewee: The operation will involve taking out loans, paying them back, and then providing loans to others. By doing this, we earn interest, which will lead to an increase in funds. The same money must be circulated continuously."
>
> -IDI, P14, Male, Leprosy affected

> "We should learn. We should not include people who do not repay loans. We should keep trustworthy local staff. Otherwise, our hardship will go in vain."
>
> -IDI, P18, Female, Marginalized

Despite the progress, some of the facilitators observed initial skepticism about savings contributions due to members past experiences with fraud in different financial institutions. Disbursement of limited loan amounts only for small investments due to limited funds led to dissatisfaction and some members wanting to leave the group. Additionally, local government skepticism about cooperative registration stemming from experiences with transient organizations posed challenges, requiring facilitators to build trust through persistent efforts. When funds or grants were limited, only a portion of the members would receive help, causing complaints and unrest among those who did not benefit.

> "Many started complaining about the savings amount, 'We don't have money with us. Many banks (financial institutions) came, took our deposits, and then disappeared; this might be the same'. The main problem is that there are many types of microfinance institutions which lend as much money as members want to borrow. However, in our group, we don't have that much money."
>
> -KII, LFA 02, Male, Facilitator

Concerns were raised about the insufficient coordination. Despite claims of cooperation, there was a noticeable gap in communication and effective collaboration, particularly in securing promised seed capital and substantial aid, despite verbal support from the local municipality office.

Effective leadership, financial discipline, and continuous capacity building were considered essential to the cooperatives sustainability. The facilitators emphasized regular group meetings, with a structured agenda focusing on savings collection, loan disbursement, and discussion of relevant topics like training and personal hygiene. Additionally, members adhered to robust financial management practices, including clear procedures for loans, interest rates, and penalties for late payments.

## 4. Sustainability

### Community and family involvement in leprosy management

Some of the stakeholders emphasized the prevention of complications from leprosy, particularly the development of deformities and ulcers. They stressed that merely educating and treating leprosy patients was insufficient; family members needed to be involved, informed, and engaged. This integrated approach would foster better care and support for individuals affected by leprosy. They believed that family involvement was critical for the rehabilitation of leprosy patients.

Stakeholders further suggested active advocacy from organizations to promote the message that leprosy was not a highly contagious disease, and that people affected by it had the right to live peacefully. Further, they suggested engaging local health posts in public awareness programs and regular health check-ups to address fear and misinformation about leprosy.

> *"To prevent further complications of leprosy, it is important not only to treat and raise awareness among patients, but also to educate their family members about the treatment process, necessary care, and assistive devices. Programs should involve both patients and their families, as family play a vital role in supporting their loved ones."*

-KII, Broader 01, Male, Health Coordinator

Most stakeholders noted that collaboration between government bodies and organizations, such as the Leprosy Mission, had effectively raised awareness. Programs on World Leprosy Day, focusing on self-care and the distribution of assistive devices, had a positive impact. Some mentioned that the ward and municipal offices have been conducting annual health and social issue screening programs for students. They stressed that public awareness initiatives have played a crucial role in reducing stigma and encouraging leprosy patients to come forward for treatment. At the same time, outreach programs have helped in identifying and addressing hidden cases of leprosy.

However, budget constraints limit the scope and frequency of such programs, revealing municipal resource allocation gap. Many stakeholders emphasized that the stigma toward leprosy remained prevalent in rural and hilly regions due to limited education, low awareness, and inadequate initiatives.

Some of the stakeholders highlighted the complexities in managing leprosy treatment and the significant role of maintaining confidentiality. Confidentiality has been crucial for treating leprosy patients due to the stigma, with some patients preferring to change their names and addresses while receiving treatment. The persistent social stigma associated with leprosy continued to affect patients' willingness to seek treatment and their social integration.

## Discussion

The project interventions significantly enhanced participants' behaviour, confidence, and social integration by strengthening local economies, promoting better hygiene and self-care, and raising awareness about leprosy as a curable disease. These efforts contributed to a reduction in stigma–a persistent challenge for individuals affected by leprosy, especially in marginalized communities. Training in healthcare, agriculture, disaster management, and financial support improved economic stability, although challenges like unequal access to training access and limited resources remained.

The transition to cooperatives aimed at long-term sustainability, but it faced challenges such as uncertainties of local government support and loan management issues, emphasizing the need for unity and trust.

Support from self-help groups emerged as a vital element in managing the challenges associated with leprosy. These groups not only facilitated access to leprosy care but also provided education on health and hygiene. Participants gained essential knowledge on early detection, the curability of leprosy, and the importance of treatment adherence and self-care practices, such as hygiene and wound care. This is consistent with findings from a study in rural Nepal that examined a group-based approach to stigma reduction among people affected by leprosy, which reported improved health literacy among self-help group members [15]. Similarly, another study assessing women's empowerment in mixed self-help groups found that women involved in SHGs demonstrated significantly higher levels of self-efficacy [16]. Prior to joining self-care groups, many individuals affected by leprosy faced challenges in accessing appropriate healthcare, including experiences of misdiagnoses, lengthy travel to health centres, and emotional distress marked by fear and anxiety surrounding their diagnosis, which aligns with earlier research involving leprosy affected patients and their careers, which found that limited knowledge about the disease contribute to delayed care seeking, heightened stigma, and significant psychological and financial burden on patients [17]. Additional findings indicated misdiagnoses and poor contact tracing

adversely affected treatment outcomes [18]. By promoting self-care, the groups empowered individuals to take control of their health, thus reducing their dependency on external healthcare providers and enhancing their confidence in managing their condition independently.

The transition of self-care groups to self-help groups not only helped participants recognize the risks associated with untreated leprosy but also served as a crucial platform to find solidarity. This shift expanded group membership to include people living with disabilities, those affected by lymphatic filariasis, and single women. In this context, self-help groups became a supportive environment for marginalized individuals, fostering inclusion and addressing broader social and economic challenges. This finding is consistent with study conducted on effectiveness of self-care groups in Ethiopia where beneficiaries had a positive attitude towards the various aspects of the programme [19].

The SHG's educational initiatives dispelling misconceptions about leprosy and contributed to a more inclusive environment. This aligns with the study a study conducted in Nepal, which aimed to enhance the mental well-being of individuals affected by leprosy. The study found that self-help groups promoted a sense of community by uniting people regardless of their background, gender, age, or health condition [20]. Another study conducted in Nigeria showed that vocational training were a valuable elements in reducing stigma protecting individuals against the loss of social value, and by facilitating their continued engagement in daily social roles in the family/community [21]. Similarly, another study in Nigeria showed blend of stigma reduction interventions include treatment, psychological and social support from families, and poverty reduction initiatives and involvement in community development initiatives [22]. Along with it such initiatives had a remarkable effect on improving patients' knowledge, practice level, patient attitude, and health consequences of leprosy [23,24].

The SHG's efforts have contributed to a more inclusive environment, consistent with prior studies on the lived experiences of leprosy-affected individuals in Nepal, Nigeria and Brazil [22,25–27]. Additionally, the project fosters community ownership of health management, which is vital for sustainability. This empowerment extends to economic opportunities via cooperatives, contributing to long-term financial independence for marginalized individuals.

The psychological impact of leprosy is profound, with many individuals reporting feelings of hopelessness and fear of disability. Social isolation, often reinforced by lack of support from family and friends, further reinforcing their sense of isolation. The findings of this study align with previous research that highlights the multifaceted impact of leprosy-related stigma on individuals. Similar to our results, a study examining the effects of stigma found significant influences on emotions, thought patterns, behavior, and interpersonal relationships [28]. These psychosocial effects often extend beyond the immediate experience of the disease, shaping how individuals perceive themselves and interact with others. A comprehensive review further supported this, emphasizing that people coping mechanisms in response to leprosy are largely shaped by their personal interpretation of the disease and its treatment [29]. Additionally, depressive and anxiety disorders were reported among persons affected by leprosy. Emotional responses such as fear, shame, and low self-esteem are commonly experienced, further compounding the psychological burden of the disease [30]. However, family support and reassurance regarding the curability of leprosy have been crucial in alleviating these concerns. A line of research indicates that leprosy patients with strong family support experience an improved quality of life [31–33].

These findings contribute to broader global health discussions by demonstrating that healthcare interventions, especially for diseases like leprosy, must go beyond medical treatment. In contexts of poverty and social exclusion, integrating social support structures (e.g., self-help groups) with economic empowerment (cooperative models) creates a framework for sustained recovery. The transformative nature of SHGs in this context aligns with multiple studies, which depict the idea that SHGs contributed a lot of progressive changes in lives of women after joining SHGs in different parts of India [34–37]. The empowerment to stand up for oneself and others represents a shift from passive acceptance of marginalization to active participation in seeking justice and pursuing economic opportunities. This empowerment not only enhance economic opportunities but also foster critical social and legal empowerment as seen in a case studies in India [38] promoting economic empowerment independence, self confidence, social cohesion, and serving an effective role towards women empowerment, social solidarity [39,40]. but.

Beyond the economic and social impacts, the interventions profoundly influenced participants' self-esteem. Many group members, initially hesitant to engage in public speaking or participate in social activities, gained the confidence to voice their concerns, share experiences, and seek support from external sources. This marks a notable shift in their social integration, as participants who had previously felt marginalized become more actively engaged in their communities. These findings align with studies that assessed the impact of self-care initiatives and program designed to address the issue of leprosy stigma in Nepal [16,41] and resonate with similar outcomes observed in self-care groups across the globe. The development of self-care groups for leprosy affected people in Ethiopia, South Sulawesi, Brazil and India led to increased confidence to participate in society, restored dignity and self-respect, and a sense of belonging within the community [2,42–44].

The transition from SHGs to a cooperative model represents a significant shift in the groups structure and approach to economic sustainability. The cooperative model offers a formal, legally backed framework for savings, loans, and financial management, addressing concerns about the sustainability of informal SHGs. This transition supports the broader argument that the cooperatives, with their governance structures and financial security offer greater long-term sustainability compared to informal self-help groups. Evidence from a case study in Karnataka, India, supports this perspective, showing that cooperatives with stable leadership tend to deliver better financial outcomes for their members [45].

Leadership and financial discipline emerged as key factors in the effective functioning of the cooperative. Structured agenda, regular meetings, and clear loan procedures highlighted the importance of strong governance which resonates with studies conducted in Nepal and other countries on ensuring the sustainability of cooperatives and other community-based organizations [46,47]. However, challenges such a lack of external support, limited coordination, and insufficient business skills pose vulnerabilities to the cooperative's success. These challenge align with research on microfinance issues in Nepal, which underscores the need for improved financial literacy and coordination [48].

Notably, individuals who have recovered from leprosy demonstrated a strong commitment to community support, actively helping others affected by the disease in seeking medical help and raise awareness. Many took roles in raising awareness, encouraging others to seek treatment, and providing personal support, particularly to those hesitant to consult healthcare professionals. This positive shift in mindset reflects increased confidence and a desire to contribute meaningfully to their communities. This phenomenon resonates with findings from studies conducted in Nepal suggesting that self-help groups can empower individuals and redirect their focus toward community development activities [20,41].

In the context of Nepal, the study highlights the critical importance of community-led interventions. Despite Nepal achieving leprosy elimination as a public health concern, there remain significant disparities in health access, particularly for those living in rural and impoverished regions. This intervention, rooted in self-care practices and local ownership, bridges the gap between policy declarations and on-the-ground realities, ensuring that marginalized populations have the tools to manage their health even in the absence of external support. The practice of pooling small contributions illustrates that even minimal inputs can accumulate into significant funds, fostering a sense of shared responsibility among members. Additionally, the expectation to repay borrowed amounts reflects a culture of accountability and trust within the cooperative. The combination of financial mechanisms, social cohesion, and a culture of accountability creates a strong foundation for the cooperative's ongoing sustainability.

A limitation of our study is that the in-depth interviews participants were selected using a random sampling technique, which may not have adequately captured the lived experiences. Additionally, since the interviews were not conducted in the local language, there is possibility of misinterpretation or limited depth of responses. As the people from the region speaks local languages such as Bhojpuri, Awadhi, Tharu and Maithili to lesser extent but it was resolved by hiring local project officer.

## Conclusion

The study highlights that self-care practices, self-help groups, and the transition to cooperative models directly contributed to improve health behaviour, reduced stigma and increased economic resilience among people affected by leprosy. The

qualitative data- drawn from focus groups, in-depth interviews, and key informant interviews of participants capture lived experiences and reflect on the role of self-care practices, SHGs, and the transition to cooperative models in fostering empowerment, improving health outcomes, and reducing stigma. SHGs played an important role in promoting social inclusion, health education, and community-based support for person affected with leprosy and related disabilities. Economic empowerment through self-help groups and the transition to a cooperative model are helpful to start businesses, improve financial stability, and support families. This has clear public health relevance, offering a scalable model that aligns the WHO's post-elimination goals and supports long-term disability-inclusive leprosy control.

## Supporting information

**S1 File. Ethics approval document.** Includes ethical approval letter from Nepal Health Research Council and ethics clearance details for the study (NHRC 444–2020 P).
(PDF)

**S2 File. COREQ checklist.** Consolidated criteria for reporting qualitative research (COREQ) outlining adherence to methodological and reporting standards for qualitative studies.
(PDF)

**S3 File. Codebook.**
(XLSX)

## Acknowledgments

We would like to extend our heartfelt gratitude to the individuals and organizations that made this study possible. Our sincere thanks go to the dedicated healthcare professionals, community health workers, and local leaders in Nepal, whose insights and support were invaluable to our research. We are especially grateful to the members of the cooperative and other participants who shared their experiences, allowing us to gain a deeper understanding of the impact of leprosy elimination efforts on their lives. Furthermore, we acknowledge Professor Frances Griffiths, Professor Richard Lilford, Sopna Choudhury, Ona ilozumba, and Jo sartori for their invaluable support throughout the manuscript writing process and support during study. We also acknowledge the contributions of the NIHR RIGHT Grant SHERPA study team and IMPACT project team as well. We are very grateful to the participants who so generously took part in this study.

## Author contributions

**Conceptualization:** Dilip Shrestha, Indra Bahadur Napit.

**Data curation:** Dilip Shrestha, Indra Bahadur Napit.

**Formal analysis:** Bijaya Shrestha.

**Funding acquisition:** Jemish Acharya, Shovakhar Kandel, Indra Bahadur Napit.

**Investigation:** Dilip Shrestha, Indra Bahadur Napit.

**Methodology:** Dilip Shrestha, Indra Bahadur Napit.

**Project administration:** Dilip Shrestha, Subi Ansari, Sangeet Sharma, Suraj Puri, Ashesh Shakya, Divya Tiwari, Samudra Pandey, Pramila Aryal, Bishnu Dhungana, Jemish Acharya, Shovakhar Kandel, Indra Bahadur Napit.

**Resources:** Indra Bahadur Napit.

**Software:** Bijaya Shrestha.

**Supervision:** Dilip Shrestha, Indra Bahadur Napit.

**Validation:** Dilip Shrestha, Indra Bahadur Napit.

**Visualization:** Bijaya Shrestha.

**Writing – original draft:** Bijaya Shrestha.

**Writing – review & editing:** Dilip Shrestha, Bijaya Shrestha, Indra Bahadur Napit.

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
