## [Decision Letter · Decision Letter 0]

31 Aug 2025

Response to Reviewers
Revised Manuscript with Track Changes
Manuscript

Shaden Kamhawi

co-Editor-in-Chief

Paul Brindley

co-Editor-in-Chief

**Additional Editor Comments:**

Reviewer #2:

**Journal Requirements:**

1) Please upload all main figures as separate Figure files in .tif or .eps format. For more information about how to convert and format your figure files please see our guidelines: 

2) Some material included in your submission may be copyrighted. According to PLOSu2019s copyright policy, authors who use figures or other material (e.g., graphics, clipart, maps) from another author or copyright holder must demonstrate or obtain permission to publish this material under the Creative Commons Attribution 4.0 International (CC BY 4.0) License used by PLOS journals. Please closely review the details of PLOSu2019s copyright requirements here: PLOS Licenses and Copyright. If you need to request permissions from a copyright holder, you may use PLOS's Copyright Content Permission form.

Potential Copyright Issues:

i) Figure 1. Please (a) provide a direct link to the base layer of the map (i.e., the country or region border shape) and ensure this is also included in the figure legend; and (b) provide a link to the terms of use / license information for the base layer image or shapefile. We cannot publish proprietary or copyrighted maps (e.g. Google Maps, Mapquest) and the terms of use for your map base layer must be compatible with our CC BY 4.0 license.

3) We note that you have indicated that there are restrictions to data sharing for this study. PLOS only allows data to be available upon request if there are legal or ethical restrictions on sharing data publicly. For more information on unacceptable data access restrictions, please see https://journals.plos.org/plosntds/s/data-availability#loc-unacceptable-data-access-restrictions.

b) If there are no restrictions, please upload the minimal anonymized data set necessary to replicate your study findings to a stable, public repository and provide us with the relevant URLs, DOIs, or accession numbers. For a list of recommended repositories, please see https://journals.plos.org/plosone/s/recommended-repositories. You also have the option of uploading the data as Supporting Information files, but we would recommend depositing data directly to a data repository if possible.

4)  Please ensure that the funders and grant numbers match between the Financial Disclosure field and the Funding Information tab in your submission form. Note that the funders must be provided in the same order in both places as well.  

**Reviewers' comments:**

**Key Review Criteria Required for Acceptance?**

**Methods**

-Are the objectives of the study clearly articulated with a clear testable hypothesis stated?

-Is the study design appropriate to address the stated objectives?

-Is the population clearly described and appropriate for the hypothesis being tested?

-Is the sample size sufficient to ensure adequate power to address the hypothesis being tested?

-Were correct statistical analysis used to support conclusions?

-Are there concerns about ethical or regulatory requirements being met?

Reviewer #1: The research design/methods and approach to data analysis are clearly described. The inclusion of in-depth interviews and focus groups with people who participated in SHGs, key informant interviews with people from the communities, and interviews with self-help group facilitators (and triangulation of data from these different sources) is a strong, holistic approach to this research.

Reviewer #2: While the study provides rich descriptive material, the methodology is weakened by the absence of a clearly articulated research question or a set of specific, pre-defined objectives guiding data collection and analysis. The work unfolded over several years as part of an ongoing programme, but the design lacks a formal temporal framework or a structured plan for how observations and interviews would be sequenced, compared, or integrated over time. This long-term, open-ended engagement risks blurring the line between programme implementation and research, making it difficult to assess the consistency of data collection or the influence of evolving programme priorities on the findings. Without a clearly stated guiding question and analytical boundaries, the study functions more as an extended narrative account of programme activities and perceived impacts than as a systematically designed research investigation.

That being said, it could easily be improved by providing more transparency on the specific methodological design that was intended before the start of the study or justification for a narrative design, but with the addition of more information on when and over how much time the IDI and FGD took place.

The manuscript claims a “mixed-methods” approach but lacks meaningful integration of the quantitative aspect. The analysis of the interview transcript through content analysis does not make it a quantitative approach. Therefore, it reads like a qualitative methodology, and the mention of “mixed methods” should be removed.

Explain where and for how long you will store the collected data. For your information: The PLOS Data policy requires authors to make all data underlying the findings described in their manuscript fully available without restriction, except in cases where the data are legally or ethically restricted (for example, participant privacy is an appropriate restriction).

Limitations could be moved to the discussion section and explained in text form instead of bullet points.

Data Saturation needs elaboration. The authors briefly mention achieving saturation but provide limited detail. Explain how saturation was assessed across different participant types (e.g., SHG members vs facilitators vs stakeholders). However, the concept of data saturation in qualitative research is being debated. If the authors do not have a convincing argument for the achievement of data saturation, it can also be left out to not provide room for uncertainty.

Ethical Considerations need strengthening. While ethics approval is mentioned, there is insufficient detail on how confidentiality, translation, and interviewer–interviewee dynamics were handled in sensitive contexts. The authors could provide information if written consent was given by the interviewees after the explanations, and if not, provide a rationale. Moreover, they could expand on how interviews were conducted ethically, particularly with regard to social stigma and power imbalances.

Reviewer #3: The methods used in this study, FDG, IDI and KII are appropriate to generate qualitative information to attain to the objectives of this research. The objectives stated in the introduction seem to address the whole mixed-methods study that was conducted. Since this paper solemnly covers the qualitative part, the objectives would benefit from a more precise formulation of the research question that were relevant for the qualitative part. Formulating the questions more precise would greatly benefit the reader in understanding the relevance of the rich results and discussion this study produced. The research population and sampling strategies are described in detail, in both the “study setting” and the “sampling and participants” paragraphs. For more clarity, these two could be re-designed since information on the participants is also in the “study setting”. Data collection, saturation (incl. the sample size n=118) and analysis are clearly written and reasoned, giving a nice overview of the workflow that has been conducted. The ethical clearance was adequately obtained, and measures to ensure confidentiality and eye-level interviews are succinctly described. The only thing I am wondering is if written consent was obtained by all the participants of this study? If so, it would be good to state that explicitly in the “Ethics”. The limitations mentioned in the methods section could be placed in the “Discussion” to be discussed for the impacts on the results and further studies.

**Results**

-Does the analysis presented match the analysis plan?

-Are the results clearly and completely presented?

-Are the figures (Tables, Images) of sufficient quality for clarity?

Reviewer #1: Yes, the analysis presented matches the analysis plan in terms of how themes were identified and presented. The results and figures included are clearly presented.

Reviewer #2: The manuscript is excessively lengthy and includes redundancies, especially in the Results and Discussion sections. However, it reads nicely in a chronological way. For publication in PLOS NTDs, it is allowed to be lengthy but would benefit from a prioritisation of the most salient quotes and findings, as well as elimination of repetition of themes across sections. Instead, present some of the findings in figures that showcase the interconnectedness of themes and demonstrate the outcome. For instance, also Table 1 (if not intended already) could be presented in the main section and not restricted to the appendix. The guidelines state, “Insert tables immediately after the first paragraph in which they are cited.”

Another example, the transition from SHG to cooperative, could benefit from a visual timeline or diagram.

Although the manuscript provides rich description, it lacks engagement with social science theories of empowerment, community development, or health behaviour change.

The manuscript would benefit from the integration of basic theoretical concepts in framing the discussion. If none was used by the researchers, it can also be stated for transparency that it is solely a phenomenological observational approach and the analysis is not purposefully based on previous theoretical works.

Reviewer #3: The results descibe the successes/benefits and challenges of the measures implemented through the IMPACT project. The four key themes, self-care, SHG, cooperatives and sustainability arose from interview analysis. The results are very much presented in detail with additional context and are very well and understandably written. Under the theme of “Sustainability” the authors describe the role of family and community involvement in leprosy, and additionally the role of government actions in leprosy awareness and difficulties with leprosy awareness campaigns. This should be ideally reflected in a separate title – as it rises important new points separate from family and community involvement.

Generally, some of the interview quotes feel lengthy, and could benefit from a little shortening to emphasize and support the results better.

**Conclusions**

-Are the conclusions supported by the data presented?

-Are the limitations of analysis clearly described?

-Do the authors discuss how these data can be helpful to advance our understanding of the topic under study?

-Is public health relevance addressed?

Reviewer #1: The conclusions are well-supported by the data, and the public health relevance of the research is strong and can provide a great model for mixed self-help groups in the future. I had one suggestion for the authors to include additional information about the limitations associated with languages spoken by participants and researchers.

In the limitations section, the authors write, “Interviews were not conducted in the local language.” Could you say a little more about this? Were they in Nepali, and if so, what is the level of fluency of people for whom that is a second language or the language of school vs. what is spoken at home? What was the local language? Do you feel like this was an issue more for people in self-help group participants than for facilitators or other community leaders?

Reviewer #2: Provide a clearer statement of how this study addresses the public health relevance and a recommendation for future research.

Reviewer #3: The discussion very well summarizes the results and their significance and is very well composed. The following paragraphs discuss the implications that self-help groups have on the management of the different impacts of leprosy. It very well contextualizes the findings from the study with results from similar studies from other countries and contexts. For a more precise understanding maybe it would be beneficial to shortly outline the contexts of the studies that are mentioned, e.g. were these also studies on SHG? Were these conducted with people affected by leprosy, or other marginalized groups as well? Did you have findings that other studies did not have? Or any contradictory points? This could be worth discussing as well if there were some.

Limitations are mentioned in the methods of this manuscript. Personally, I’d suggest moving and discussing them here, as it allows to discuss the implications of the limitations for this and further research on this topic.

Overall, the discussion does very well embed the findings from this study into the broader context, emphasizing its significance and the long-term effects. The discussion part on sustainability and the need for community-led interventions demonstrates the transferability of the findings into possible other contexts as well – highlighting its relevance for public health and public health research.

**Editorial and Data Presentation Modifications?**

Reviewer #1: Here are a few minor editorial suggestions:

p. 4, lines 8-9: “Leprosy is closely linked to poverty and economic hardships because individuals with the disease often experience stigma and discrimination, beyond their physical deformities.” I would restate this to say that something about the complex relationship between leprosy and poverty, with some of the conditions of extreme poverty contributing to spread of the disease and the disease itself resulting in stigma and discrimination, which can result in or further exacerbate economic marginalization.

p. 4, lines 14-15: Explain the criteria for “elimination” as used by WHO, since it is different from the dictionary definition of the word. It might be helpful to include the latest numbers of incidence and prevalence in Nepal here too.

p. 4, line 23: Can you include a little more information about IMPACT and how it is funded? Is it an NGO?

p. 7, line 21: “facilitators”—maybe say “facilitators of the self-help groups” (or does this mean overall project facilitators)?

p. 11, line 9: “member’s” should be “members’”

p. 22: In the quote about planting sunflowers (DI, P04, Male, PLWD), is he saying the loan was not enough because it didn’t cover the cost of pesticides and fertilizers? Maybe a sentence of explanation between quotations here could be useful.

Reviewer #2: Double-check the manuscript for language and grammar issues, consulting a native English speaker or a tool.

The abstract should include specific numbers or key statistics (e.g., number of participants, number of groups).

Double-check that acronyms are clarified on their first use (e.g., PLWD).

Reviewer #3: -

**Summary and General Comments**

Reviewer #1: This is an excellent article on what seems to be a very promising model for how self-help groups can be successful and sustainable. You all do a great job demonstrating a self-help group with mixed participation from groups of different marginalized statuses and identities can help each other, and you provide a good discussion of some of the logistical challenges that were faced.

You note that a benefit of a mixed self-help group that includes people affected by leprosy is that it can help to destigmatize leprosy. I found this also in observing an existing discussion group that was held in a general dermatology treatment center in Brazil; there were people coming in for many other conditions, but a social worker would lead discussions about leprosy, and this led to people bringing up their misconceptions and learning new things. You mentioned that more affluent or higher caste people affected by leprosy may not want to attend a group that is leprosy-specific, but were they more likely to attend when the group was expanded to include other marginalized groups?

Reviewer #2: Dear authors of PNTD-D-25-00023,

Thank you for a highly relevant and original study on the transition from Self-Help Groups (SHGs) to cooperatives within the context of leprosy-affected and marginalised communities in Nepal under the IMPACT initiative. The manuscript addresses an important and under-researched issue in neglected tropical diseases (NTDs), particularly the intersection of social empowerment, stigma, and sustainability in leprosy-affected communities. The shift from SHGs to cooperatives is a novel angle, showing a long-term development trajectory beyond health.

The study design includes focus group discussions (FGDs), in-depth interviews (IDIs), key informant interviews (KIIs), and observation, providing data triangulation and depth. Moreover, the use of COREQ guidelines demonstrates rigour and transparency in qualitative research.

The narrative quotes provide powerful, real-life illustrations of transformation in the lives of marginalised individuals. The cultural and gender-sensitive findings are well-articulated.

Reviewer #3: Introduction: The introduction gives a clear outline on the topic of leprosy in Nepal, the sociocultural implications of leprosy and the role of self-help and -care. The topic of the paper is well introduced. Moreover, it introduces the reader to the IMPACT study. This part of the introduction could benefit from some examples on what the interventions within the IMPACT project look like to better contextualize the findings from the results. For additional context, it might be beneficial to embed the study in a summary of some pre-existing literature, that seem highly relevant or even linked to the study, such as Pryce et al. 2018 (Assessing the feasibility of integration of self-care for filarial lymphoedema into existing community leprosy self-help groups in Nepal) or Shresta et al. 2021 (Evaluation of a self-help intervention to promote the health and wellbeing of marginalized people including those living with leprosy in Nepal: a prospective, observational, cluster-based, cohort study with controls).

General Comments: The findings of this study are very interesting, and seem to depict an innovative, sustainable and long-term approach to address marginalization and stigmatization of people affected by leprosy and other marginalized groups. Overall, the manuscript offers new insights relevant to researchers and policy makers not only in Nepal but also in other settings. The main points to address before publication revolve around the clearance of the objectives and research questions, to provide the reader with a clear outline of this research, and the contextualization of the research itself (in the introduction) and the contextualization of the results in the discussion part. Considering the results describe successes and challenges of the measures implemented in the IMPACT project – maybe this should reflect in the title (and the research question)?

PLOS authors have the option to publish the peer review history of their article (what does this mean? ). If published, this will include your full peer review and any attached files.

**Do you want your identity to be public for this peer review?** For information about this choice, including consent withdrawal, please see our Privacy Policy .

Reviewer #1: **Yes: ** Cassandra White

Reviewer #2: No

Reviewer #3: No

**Figure resubmission:**

**Reproducibility:** To enhance the reproducibility of your results, we recommend that authors of applicable studies deposit laboratory protocols in protocols.io, where a protocol can be assigned its own identifier (DOI) such that it can be cited independently in the future. Additionally, PLOS ONE offers an option to publish peer-reviewed clinical study protocols. Read more information on sharing protocols at https://plos.org/protocols?utm_medium=editorial-email&utm_source=authorletters&utm_campaign=protocols

---

## [Decision Letter · Decision Letter 1]

30 Oct 2025

Response to Reviewers
Revised Manuscript with Track Changes
Manuscript

Kind regards,

Shaden Kamhawi

co-Editor-in-Chief

Paul Brindley

co-Editor-in-Chief

**Journal Requirements:**

**Reviewers' comments:**

**Key Review Criteria Required for Acceptance?**

**Methods**

-Are the objectives of the study clearly articulated with a clear testable hypothesis stated?

-Is the study design appropriate to address the stated objectives?

-Is the population clearly described and appropriate for the hypothesis being tested?

-Is the sample size sufficient to ensure adequate power to address the hypothesis being tested?

-Were correct statistical analysis used to support conclusions?

-Are there concerns about ethical or regulatory requirements being met?

Reviewer #2: Clarity of Research Question:

The Introduction is much improved, but please include a clear, single-sentence research question or aim at the end of the section to sharpen the study’s focus.

Study Design Terminology:

The manuscript is primarily qualitative, though part of a larger mixed-methods study. Consider rephrasing to emphasize that this paper reports the qualitative component of that broader design.

Data Saturation Description:

Clarify how data saturation was determined (e.g., no new codes/themes emerged after a certain number of interviews or groups).

Reviewer #3: To improve quality, I would suggest avoiding explaining the mixed-methods approach, since the presented research in this article is solemnly qualitative.

The WHO methodology used for the interviews needs referencing.

Reviewer #4: The Introduction and Methods part of the abstract do not clearly enough correspond to the qualitative study described in the article but rather refer to the broader IMPACT initiative. This could mislead the reader. Please revise the abstract to reflect better the qualitative component and clearly distinguish between the overall IMPACT programme and this specific study.

Similarly, the objective appears to be linked to the full IMPACT study. Objectives referring to the qualitative study should be stated especially considering that a clear, guiding research question is missing.

The introduction could be strengthened by a more thorough discussion of the existing literature and an explicit emphasis on what is unknown and what gap you want to fill. What question are you answering? Reading the title, one expects something along the lines of ‘What are the experiences and perceptions of key stakeholders (people affected by leprosy, facilitators, community leaders...) regarding the IMPACT initiative to sustain community-based leprosy programmes’. Later, it sounds more like: "What are the experiences and perceptions of key stakeholders in the IMPACT initiative (people affected by leprosy, facilitators, community leaders...) regarding the successes and challenges of the IMPACT programme (the role of self-care, participation in self-help groups) in promoting health equity, disease control and socio-economic empowerment, as well as cooperative transformation in sustaining community-based leprosy programmes?" The guiding research question should be clear and consistent from the title to the conclusion.

Methods

Although most of the important information is included, the Method section would benefit from some of the Infromation being reorganised and allocated more fittingly into the different subsections.

Study design - P 5: Consider starting the paragraph of “study design” with “The larger IMPACT study employed a mixed-method design…..” to make the delineation between full IMPACT and this qualitative study clearer.

Study setting - Quite clear. The differentiation between the IMPACT programme implementation area and the sub-areas considered in this study could be a bit clearer. The Description of study population belongs in the participant section

Participants and Sampling – What were the inclusion and exclusion criteria, the target group and the reasons for the sampling strategy. The rationale for using a random rather than a purposive sampling technique should be explained. The description of data collection methods (IDI, KII…) belongs to the data collection section. On page 6 you mention “a checklist based on the framework to record behaviour change in this section”– what do you mean? Is this a theoretical framework that guided methodology/analysis? If so this would make more sense in the introduction and/or study design.

Data collection - Summarise how, when, and by whom the data was collected, including the instruments (interview guides), duration, and logistics of the sessions. Many parts currently mentioned in the participant section would be better placed here.

Data analysis - This section is clear.

Ethical considerations - This section needs to be strengthened.

1. You mention that measures were taken to ensure that people felt comfortable (p 8) – what were these measures? As some of the participants belong to vulnerable groups please include that measures to minimise harm and ensure psychological/emotional safety were taken, for example, accessible venues for FGDs and interviews, as some participants had disabilities.

2. Voluntary participation and voluntary withdrawal should be explicitly mentioned, as should measures to avoid coercion – especially as this statement “They described their efforts to convince people to join these groups despite facing significant resistance, emphasizing a personal commitment to social change” (p 13) raises questions about social coercion. What were the efforts to persuade people to participate, and how did you avoid coercion?

**Results**

-Does the analysis presented match the analysis plan?

-Are the results clearly and completely presented?

-Are the figures (Tables, Images) of sufficient quality for clarity?

Reviewer #2: Conciseness of Results:

he Results section is rich in participant narratives but remains lengthy. Some repetitive quotations could be reduced without losing the depth of insight. Aim for clearer synthesis across the four major themes.

Reviewer #3: (No Response)

Reviewer #4: The results are comprehensive, rich in qualitative detail and underpinned by well-chosen quotes from participants. However, the absence of a clearly formulated research question or guiding framework sometimes makes it difficult for the reader to follow the thread.

Regarding figures:

- Page 7: you refer to Supplementary table 1) for information on the 80 participants participating in the FGDs. Consider also adding the characteristics of the FDG participants (gender, age, condition etc.) to Table 1 for clarification and completeness.

- The map is currently not sufficient in quality. The font is too small to be legible and cannot really be assessed.

- Supplementary Table 1 lists a 106-year-old participant - please verify that this information is correct.

**Conclusions**

-Are the conclusions supported by the data presented?

-Are the limitations of analysis clearly described?

-Do the authors discuss how these data can be helpful to advance our understanding of the topic under study?

-Is public health relevance addressed?

Reviewer #2: (No Response)

Reviewer #3: (No Response)

Reviewer #4: The conclusions are consistent with the results.

The phrasing “transition of marginalized people into sustainable groups” is somewhat odd. A more apt formulation might be something like “The study highlights the perspectives and experiences of key actors within the IMPACT initiative on how marginalized individuals — particularly persons affected by leprosy — may move from social exclusion to empowerment through participation in self-help groups, and how these groups can transition into sustainable cooperatives.”

The finding - that the inclusion of other marginalised individuals is very important in order to reduce stigmatisation and remove barriers to joining the group - is key and could have been emphasised even more strongly.

Consider including as a limitation the exclusion of non-participants, as those who declined or dropped out may have offered important contrasting insights.

The public health relevance of community-driven, cooperative-based models for sustainable leprosy management becomes very clear.

**Editorial and Data Presentation Modifications?**

Reviewer #2: Language and Style:

A final light language edit is recommended to correct minor grammatical inconsistencies and streamline phrasing for readability.

Figures and Supplementary Material:

Ensure all figures, tables, and supplementary files comply with PLOS formatting and are properly referenced in the text.

Reviewer #3: (No Response)

Reviewer #4: P 10/ Line 4: Replace "patient" information brochure with "participant" information.

Avoid using the terms beneficiaries and patients. Instead, refer to SHG members as participants or group members to acknowledge their active roles in the initiative and their equal status as knowledge bearers and contributors to the research process.

A few minor linguistic errors remain, please review and revise accordingly.

**Summary and General Comments**

Reviewer #2: Dear authors of PNTD-D-25-00023,

Thank you for your immense work after receiving the first round of feedback. It is a very valuable contribution to meet research needs and should be published. Allow me to highlight a few areas for improvement to work on a further improved quality of the manuscript, which underlines the importance of such a study.

The authors have addressed nearly all of the major concerns raised in the previous review. The revised manuscript demonstrates substantial improvement in structure, methodological transparency, and analytical depth. The objectives are now clearly stated, the qualitative design and data analysis are well described, and ethical and contextual considerations are properly articulated. The organization of results under four key themes (self-care, SHG, cooperatives, and sustainability) provides coherence and effectively links participant narratives to the broader discussion of empowerment and stigma reduction. The Discussion section now draws meaningfully on international literature and situates the findings within global NTD and community health frameworks.

Minor issues remain: the paper still reads as largely qualitative and should be framed as such rather than a full mixed-methods study. Some redundancy in the results and minor grammatical inconsistencies could be reduced for clarity. Adding a single explicit research question in the Introduction would strengthen focus.

Overall, the manuscript now meets publication standards for PLOS NTDs with only minor editorial adjustments.

Reviewer #3: The manuscript and its clarity would benefit from a specific concrete research question in the introduction, that was the foundation for the interviews conducted.

Maybe the limitations in the discussion section can be somewhat indicated by a headline, to not weaken the well-made points above.

Reviewer #4: This is a highly relevant study with an innovative angle that fills an important knowledge gap in the field of community-based care. Specifically, it addresses the barriers and enablers to a successful transition from self-help groups to cooperative models for greater sustainability and inclusion of people with lived experience of leprosy and/or living with disabilities.

The idea of establishing a cooperative is promising, but it would have been helpful to define the term ‘cooperative’ more clearly in the introduction.

The strengths of the article lie in its rich qualitative data, its strong community focus, and its practical implications for public health and social inclusion.

The following areas could be improved:

Clarification of the objective and research question and its consistent alignment in the title, introduction, methods, and conclusion.

Improving the focus and structure of the ‘Methods’ section – among other things by distinguishing between the broader IMPACT initiative and the qualitative component and strengthening the ‘Ethical considerations’ section.

PLOS authors have the option to publish the peer review history of their article (what does this mean? ). If published, this will include your full peer review and any attached files.

**Do you want your identity to be public for this peer review?** For information about this choice, including consent withdrawal, please see our Privacy Policy .

Reviewer #2: No

Reviewer #3: No

Reviewer #4: No

**Figure resubmission:**

**Reproducibility:** To enhance the reproducibility of your results, we recommend that authors of applicable studies deposit laboratory protocols in protocols.io, where a protocol can be assigned its own identifier (DOI) such that it can be cited independently in the future. Additionally, PLOS ONE offers an option to publish peer-reviewed clinical study protocols. Read more information on sharing protocols at https://plos.org/protocols?utm_medium=editorial-email&utm_source=authorletters&utm_campaign=protocols

---

## [Decision Letter · Decision Letter 2]

26 Nov 2025

Dear Mr Shrestha,

We are pleased to inform you that your manuscript 'Fostering Empowerment: Transition from Self-Help Groups to Cooperatives in Leprosy-Affected Communities in Nepal' has been provisionally accepted for publication in PLOS Neglected Tropical Diseases.

Best regards,

Anil Fastenau, M.D., M.Sc.

Guest Editor

Sitara Ajjampur

Section Editor

Shaden Kamhawi

co-Editor-in-Chief

Paul Brindley

co-Editor-in-Chief

Reviewer's Responses to Questions

**Key Review Criteria Required for Acceptance?**

**Methods**

-Are the objectives of the study clearly articulated with a clear testable hypothesis stated?

-Is the study design appropriate to address the stated objectives?

-Is the population clearly described and appropriate for the hypothesis being tested?

-Is the sample size sufficient to ensure adequate power to address the hypothesis being tested?

-Were correct statistical analysis used to support conclusions?

-Are there concerns about ethical or regulatory requirements being met?

Reviewer #2: (No Response)

Reviewer #3: While the methods section has improved, the clear structure is still missing within the text. Oftentimes, the text under the sub-headlines does not match the headline and should belong into another section. E.g. conducting KII and FGD is under participants and sampling, while it rather belongs into the data collection section. This section should be revised according to the sub-headings provided.

**Results**

-Does the analysis presented match the analysis plan?

-Are the results clearly and completely presented?

-Are the figures (Tables, Images) of sufficient quality for clarity?

Reviewer #2: (No Response)

Reviewer #3: The quotes are still lengthy, consider shortening them even more for better readability and conciseness.

**Conclusions**

-Are the conclusions supported by the data presented?

-Are the limitations of analysis clearly described?

-Do the authors discuss how these data can be helpful to advance our understanding of the topic under study?

-Is public health relevance addressed?

Reviewer #2: (No Response)

Reviewer #3: (No Response)

**Editorial and Data Presentation Modifications?**

Reviewer #2: Supply a high-resolution map with readable typography at journal layout size.

Reviewer #3: The figure with the map is still not really readable, as the font is too small.

**Summary and General Comments**

Reviewer #2: (No Response)

Reviewer #3: A clear research question, reflecting from title to conclusion is still missing.

"This research tends to understand the extent do self-care practices, self-help groups, and cooperatives improve the long-term sustainability of community-based leprosy management programs." should be revised into another research question more suitable to the results. The title should be adapted accordingly as well, making it easier to follow the research throughout the paper.

It would be supportive of the research, if the research gap could be clarified a little more using existing literature and embedding your research in it. Thus, your research question can be formulated in a clearer way as well.

PLOS authors have the option to publish the peer review history of their article (what does this mean? ). If published, this will include your full peer review and any attached files.

**Do you want your identity to be public for this peer review?** For information about this choice, including consent withdrawal, please see our Privacy Policy .

Reviewer #2: No

Reviewer #3: No

---

## [Editor Report · Acceptance letter]

Dear Mr Shrestha,

We are delighted to inform you that your manuscript, "Fostering Empowerment: Transition from Self-Help Groups to Cooperatives in Leprosy-Affected Communities in Nepal," has been formally accepted for publication in PLOS Neglected Tropical Diseases.

Best regards,

Shaden Kamhawi

co-Editor-in-Chief

Paul Brindley

co-Editor-in-Chief
